# Viral-mediated Pou5f1 (Oct4) overexpression and inhibition of Notch signaling synergistically induce neurogenic competence in mammalian Müller glia

Nguyet Le[1], Sherine Awad[2,3,4], Isabella Palazzo[1], Thanh Hoang[2,3,4], Seth Blackshaw[1,5,6,7,8]*

[1]Department of Neuroscience, Johns Hopkins University School of Medicine, Baltimore, United States; [2]Department of Ophthalmology and Visual Sciences, University of Michigan School of Medicine, Ann Arbor, United States; [3]Department of Cell and Developmental Biology, University of Michigan School of Medicine, Ann Arbor, United States; [4]Michigan Neuroscience Institute, University of Michigan School of Medicine, Ann Arbor, United States; [5]Department of Ophthalmology, Johns Hopkins University School of Medicine, Baltimore, United States; [6]Department of Neurology, Johns Hopkins University School of Medicine, Baltimore, United States; [7]Institute for Cell Engineering, Johns Hopkins University School of Medicine, Baltimore, United States; [8]Kavli Neuroscience Discovery Institute, Johns Hopkins University School of Medicine, Baltimore, United States

*For correspondence:
sblack@jhmi.edu

## eLife Assessment

This manuscript demonstrates that Oct4 overexpression synergizes with Notch inhibition (Rbpj knockout) to promote the conversion of adult murine Müller glia (MG) into bipolar cells. These findings are **important** as the authors used rigorous genetic lineage tracing (GLAST-CreER; Sun-GFP) to confirm that neurogenesis indeed originates from MGs, addressing a key issue in the field. The single-cell multiomic analyses are **compelling**, and while functional studies of MG-derived bipolar cells would strengthen the conclusions, they are beyond the scope of this study.

**Abstract** Retinal Müller glia in cold-blooded vertebrates can reprogram into neurogenic progenitors to replace neurons lost to injury, but mammals lack this ability. While recent studies have shown that transgenic overexpression of neurogenic bHLH factors and glial-specific disruption of NFI family transcription factors and Notch signaling induce neurogenic competence in mammalian Müller glia, induction of neurogenesis in wildtype glia has thus far proven elusive. Here, we report that viral-mediated overexpression of the pluripotency factor *Pou5f1 (Oct4)* induces transdifferentiation of mouse Müller glia into bipolar neurons, and synergistically stimulates glial-derived neurogenesis in parallel with Notch loss of function. Single-cell multiomic analysis shows that *Pou5f1* overexpression leads to widespread changes in gene expression and chromatin accessibility, inducing activity of both the neurogenic transcription factor Rfx4 and the Yamanaka factors Sox2 and Klf4. This study demonstrates that viral-mediated overexpression of *Pou5f1* induces neurogenic competence in adult mouse Müller glia, identifying mechanisms that could be used in cell-based therapies for treating retinal dystrophies.

## Introduction

The homeodomain factor *Pou5f1 (Oct4)* is a central component of transcriptional regulatory networks that maintain pluripotency in embryonic stem (ES) cells. Along with *Sox2*, *Klf4*, and *Myc*, *Pou5f1* is one of the Yamanaka factors sufficient to induce the formation of induced pluripotent stem cells from virtually all somatic cell types (*Takahashi and Yamanaka, 2006*). In contrast to these other factors, *Pou5f1* is selectively expressed in ES cells and essentially absent from somatic cells, including adult stem cells from every tissue examined. The two clear exceptions are found in gonadal stem cells (*Takashima et al., 2013*; *Uhlen et al., 2010*) and during neural crest formation, where *Pou5f1* expression is essential for the ability of these cells to generate diverse cell types (*Morrison and Brickman, 2006*; *Zalc et al., 2021*). This has raised the question of whether *Pou5f1* overexpression in vivo, alone or in combination with other pluripotency factors, may confer some level of multipotency on somatic cells in the central nervous system.

Sustained overexpression of the Yamanaka factors *Pou5f1*, *Sox2*, *Klf4*, and *Myc* induces teratoma formation in many tissues (*Abad et al., 2013*; *Ohnishi et al., 2014*). Transient or low-level Yamanaka factor overexpression, in contrast, has been reported to induce cellular rejuvenation in both neurons and other cell types, although this occurs without clear induction of proliferation or conferring multipotency (*Ocampo et al., 2016*; *Wang et al., 2021a*). However, several studies have directly investigated whether viral-mediated overexpression in CNS neural stem or glial cells of *Pou5f1* alone, or in combination with subsets of other pluripotency factors, can induce multipotency and neurogenesis. Results from these studies are mixed, with *Pou5f1* overexpression in neural stem cells reported to either not induce neurogenesis (*Asadi et al., 2015*; *Dehghan et al., 2015*; *Sim et al., 2011*), to promote formation of oligodendrocyte progenitors (OPCs) (*Yu et al., 2021*), or to promote remyelination in committed OPCs (*Dehghan et al., 2016*). Two studies reported that viral-mediated *Pou5f1* overexpression, either alone (*Niu et al., 2013*) or in combination with other Yamanaka factors (*Gao et al., 2016*), could directly convert reactive astrocytes into neurons. However, these studies used the GFAP minipromoter to drive gene expression, which has been reported to show insert-dependent ectopic insert-dependent silencing in glia and activation in neurons (*Wang et al., 2021b*). This limitation makes these findings difficult to interpret, as additional methods of validating cell lineage were not used.

Two previous studies have reported injury-dependent induction of *Pou5f1* in retinal Müller glial cells in zebrafish and mice. In zebrafish, where injury induces Müller glia to undergo conversion to neurogenic progenitors (*Wan and Goldman, 2016*), it was reported that *Pou5f1* is essential for glial reprogramming to occur (*Sharma et al., 2019*). In mice, where neurogenic competence is rapidly and actively suppressed following injury, it was reported that *Pou5f1* expression is transiently induced, but fully repressed within 24 hr following injury (*Reyes-Aguirre and Lamas, 2016*). This raises the possibility that *Pou5f1* expression may be sufficient to confer neurogenic competence on retinal Müller glial cells. In this study, we assessed whether AAV-mediated overexpression of *Pou5f1* could induce reprogramming of adult mouse Müller glia. While we did not detect any expression of *Pou5f1* in either zebrafish or mouse Müller glia following injury, we used both genetic cell lineage and single-cell omics analyses to unambiguously demonstrate that viral-mediated overexpression of *Pou5f1* selectively induces the formation of bipolar neurons from control GlastCreER;*Rosa26*$^{LSL-Sun1-GFP}$ mouse Müller glia without inducing substantial levels of proliferation. We further showed that *Pou5f1* overexpression synergistically enhances existing levels of Müller glia-derived neurogenesis in the absence of Notch signaling. Single-cell multiomic analysis demonstrated that *Pou5f1* cooperates with Sox2 and Klf4, which are constitutively expressed in Müller glia, to broadly alter the gene regulatory landscape and induce expression of genes such as the neurogenic transcription factor *Rfx4*. This study demonstrates that substantial levels of in situ glia-to-neuron reprogramming in mammalian retina can be induced by viral overexpression of a Yamanaka factor.

## Results

### *Pou5f1* and Nanog are not detectably expressed in Müller glia or Müller glia-derived progenitor cells in either zebrafish or mice

To examine the expression of *Pou5f1* and other pluripotency-associated genes in Müller glia, we first analyzed previously published gene expression data from zebrafish and mouse retina (*Hoang*

*et al., 2020*; *Le et al., 2024*). We first analyzed both single-cell RNA-seq data obtained from wildtype retinas and bulk RNA-seq data obtained from FACS-isolated Müller glia carrying glial-specific GFP reporter lines. Consistent with previous findings (*Nelson et al., 2011*; *Surzenko et al., 2013*), in mice we detected expression of *Sox2* and *Klf4* in resting and both *N*-methyl-D-aspartate (NMDA)-treated and light-damaged Müller glia, along with injury-induced expression of *Myc*, which is lost as activated glia progressively return to a resting state by 72 hr post-injury. Likewise, in zebrafish we observed expression of *sox2* and *myca/b* (but not *klf4*) in both resting and injured Müller glia (*Figure 1—figure supplement 1A, B*). However, we did not detect expression of either *Pou5f1* or its direct target *Nanog* in either species in any treatment condition (*Loh et al., 2006*). Bulk RNA-seq data obtained from purified Müller glia detected very low levels of *pou5f3v(Oct4)*, but not *Nanog*, expression in some injured zebrafish Muller glia, but neither *Pou5f1* nor *Nanog* was detected in either resting or injured mouse Müller glia (*Figure 1—figure supplement 1A, B*). In addition, we did not observe any induction of *Pou5f1* or *Nanog* in snRNA-seq data obtained from mouse Müller glia following selective loss of function of *Rbpj* alone or in combination with *Nfia/b/x* (*Le et al., 2024*; *Figure 1—figure supplement 1C*). These results indicate that neither NMDA excitotoxicity, light damage, or induction of neurogenic

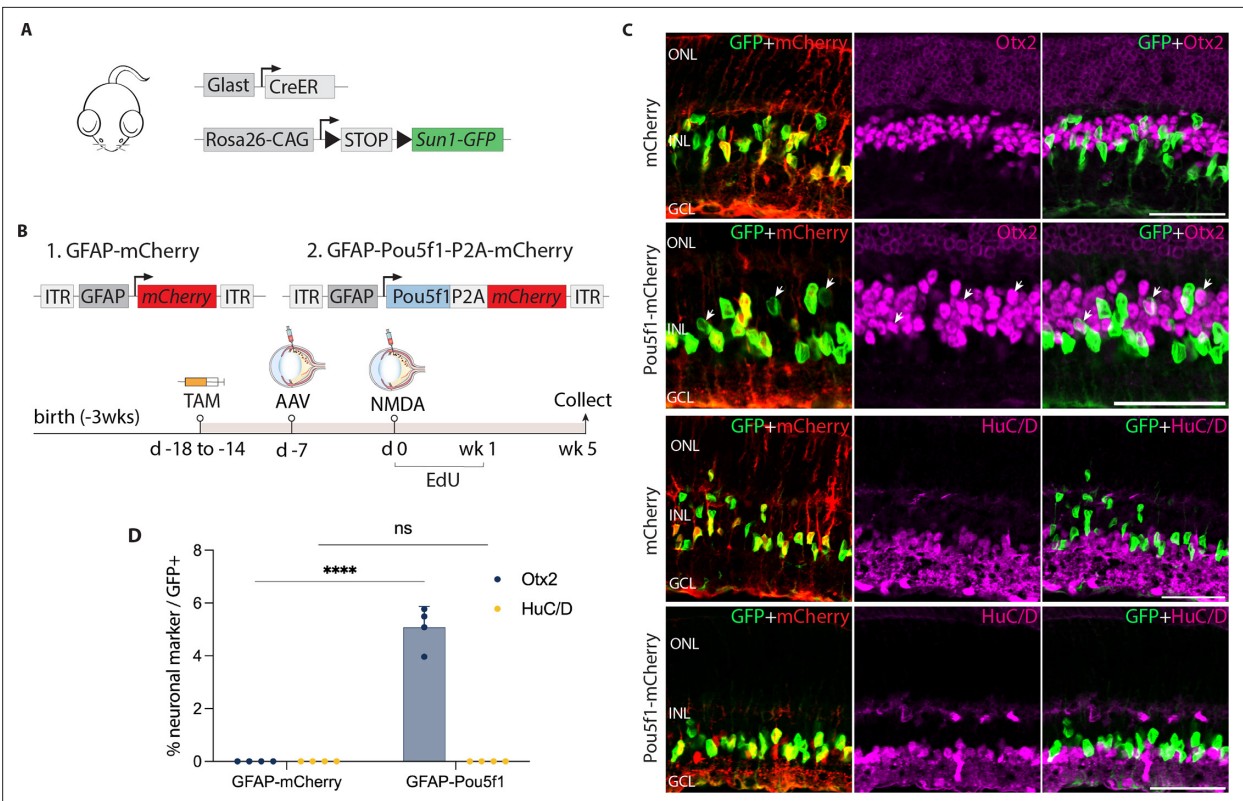

**Figure 1.** AAV-mediated overexpression of *Pou5f1* induces neurogenesis in control GlastCreER;*Rosa26^LSL-Sun1-GFP* Müller glia following *N*-methyl-D-aspartate (NMDA)-induced excitotoxicity. (**A**) Schematic of the transgenic construct used to specifically label Müller glia with Sun1-GFP expression. (**B**) Schematic of the GFAP AAV constructs and experimental workflow. (**C**) Representative images of retinas immunolabeled for GFP, mCherry, Otx2, and HuC/D. White arrowheads indicate GFP-positive Müller glia-derived neurons expressing neuronal markers Otx2 or HuC/D. (**D**) Quantification of mean percentage ± SD of GFP-positive Müller glia-derived neurons expressing either Otx2 or HuC/D. Significance was determined via two-way ANOVA with Tukey's multiple comparison test: ****p < 0.0001. Each data point was calculated from an individual retina. TAM, tamoxifen; ONL, outer nuclear layer; INL, inner nuclear layer; GCL, ganglion cell layer. Scale bar = 50 μm.

The online version of this article includes the following figure supplement(s) for figure 1:

**Figure supplement 1.** *Oct4/pou5f3* mRNA is not detectably expressed in Müller glia after injury.

**Figure supplement 2.** Immunohistochemical analysis of glial markers, AAV reporter expression, and other molecular markers at 1 and 2 weeks following *N*-methyl-D-aspartate (NMDA) in control GlastCreER;*Rosa26^LSL-Sun1-GFP* Müller glia.

**Figure supplement 3.** Immunohistochemical analysis of glial markers, AAV reporter expression, and other molecular markers at 5 weeks following *N*-methyl-D-aspartate (NMDA) in control Müller glia.

competence by loss of function of *Nfia/b/x* and/or *Rbpj* induces detectable levels of *Pou5f1* or *Nanog* in neurogenic mammalian Müller glia.

## AAV-mediated overexpression of *Pou5f1* induces conversion of control GlastCreER;*Rosa26^LSL-Sun1-GFP* mouse Müller glia to bipolar-like cells

To determine whether *Pou5f1* overexpression could reprogram mammalian Müller glia into a neurogenic state, we administered intraperitoneal tamoxifen injections into P21 GlastCreER;*Rosa26^LSL-Sun1-GFP* mice, thereby irreversibly labeling Müller glia and their progeny with nuclear envelope-targeted *Rosa26^LSL-Sun1-GFP* under control of the ubiquitous CAG promoter (*de Melo et al., 2012*; *Hoang et al., 2020*; *Figure 1A*, *Figure 1—figure supplement 2a*). One week following the final tamoxifen dose, we performed intravitreal injections of either Gfap-*Pou5f1*-mCherry or Gfap-mCherry control 7m8 AAV (*Figure 1B*, *Figure 1—figure supplement 2b*). While expression of other neurogenic bHLH factors driven by the GFAP promoter were rapidly silenced in Müller glia and activated in amacrine and retinal ganglion cells, Gfap-*Pou5f1*-mCherry remained selectively expressed in Müller glia. However, *Pou5f1* expression alone did not induce detectable levels of Müller glia-derived neurogenesis in the uninjured retina (*Le et al., 2022*).

Since injury is often required to promote Müller glia-derived neurogenesis following *Ascl1* overexpression or loss of function of *Nfia/b/x* (*Hoang et al., 2020*; *Jorstad et al., 2017*), we hypothesized that this might also be so for Müller glia overexpressing *Pou5f1*. To test this, we injected NMDA 1 week post-viral injection. We then administered 5-ethynyl-2'-deoxyuridine (EdU) to label proliferating cells. Retinas were harvested and analyzed at 1, 2, and 5 weeks post-injury (*Figure 1—figure supplement 2b*, *Figure 1B*). Immunostaining analysis revealed enriched mCherry expression in GlastCreER;*Rosa26^LSL-Sun1-GFP* Müller glia in both control mCherry and *Pou5f1*-mCherry-infected retinas at all timepoints (*Figure 1C*, *Figure 1—figure supplement 2c–l*). In addition, *Pou5f1* expression was detected in GFP-positive Müller glia in *Pou5f1*-mCherry-infected retinas but not in mCherry-infected GlastCreER;*Rosa26^LSL-Sun1-GFP* retina (*Figure 1—figure supplement 2c, d*, *Figure 1—figure supplement 3*). While most GFP-positive cells infected with *Pou5f1*-mCherry expressed the Müller glial marker Sox2 (*Figure 1—figure supplement 2e, f*, *Figure 1—figure supplement 3b*), a subset of GFP-positive cells lost Sox2 expression from 2 weeks post-injury onward (*Figure 1—figure supplement 2f*, *Figure 1—figure supplement 3b*, yellow arrows), which is consistent with glia-to-neuron conversion.

We observed selective expression of the neurogenic bHLH factor Ascl1 in *Pou5f1*-mCherry-infected GFP-positive Müller glia at 1 week post-injury (*Figure 1—figure supplement 2g*), with the number of Ascl1-positive cells declining somewhat by 2 and 5 weeks post-injury (*Figure 1—figure supplement 2h*, *Figure 1—figure supplement 3c*). While at 1 week post-injury, we did not observe any significant Müller glia-derived neurogenesis (*Figure 1—figure supplement 2i, m*), by 2 weeks post-injury we observed 1.1% of GFP-positive cells colocalized with the bipolar cell marker Otx2 in *Pou5f1*-mCherry-infected retinas (*Figure 1—figure supplement 2j, m*), and this increased to 5.1% of GFP-positive cells at 5 weeks post-injury (*Figure 1A, D*). We found that Müller glia-derived Otx2+ bipolar-like cells lost mCherry reporter expression, likely due to reduced activity of the Gfap-*Pou5f1*-mCherry construct in neurons (*Le et al., 2022*). Immunostaining with additional cell-specific markers revealed that a subset of these GFP-positive cells expressed the cone bipolar markers Scgn and Cabp5 (*Figure 1—figure supplement 3d, e*), but not the rod and cone ON bipolar cell marker Isl1 or the rod marker Nrl (*Figure 1—figure supplement 3f, g*). Furthermore, we did not observe expression of the amacrine cell marker HuC/D in GFP-positive cells (*Figure 1C, D*), which was observed in neurons generated from neurogenic mouse Müller glia deficient in either *Rbpj* or *Nfia/b/x* (*Hoang et al., 2020*; *Le et al., 2024*). No EdU/GFP-positive cells were detected at either 1 week (*Figure 1—figure supplement 2k*) or 5 weeks post-injury (*Figure 1—figure supplement 3h*). However, at 2 weeks post-injury, we observed a very limited number of GFP-positive cells incorporating EdU in retinas infected with both mCherry control and *Pou5f1*-mCherry, with modest EdU incorporation but significantly increased by *Pou5f1* overexpression (*Figure 1—figure supplement 2l, n*). These findings indicate that *Pou5f1* overexpression induces a very low level of proliferation in Müller glia in the second week following NMDA injury. However, the absence of EdU/Otx2/GFP-positive cells at both 2 and 5 weeks post-injury suggests that bipolar cell generation at this stage was, at least in part, driven by direct transdifferentiation.

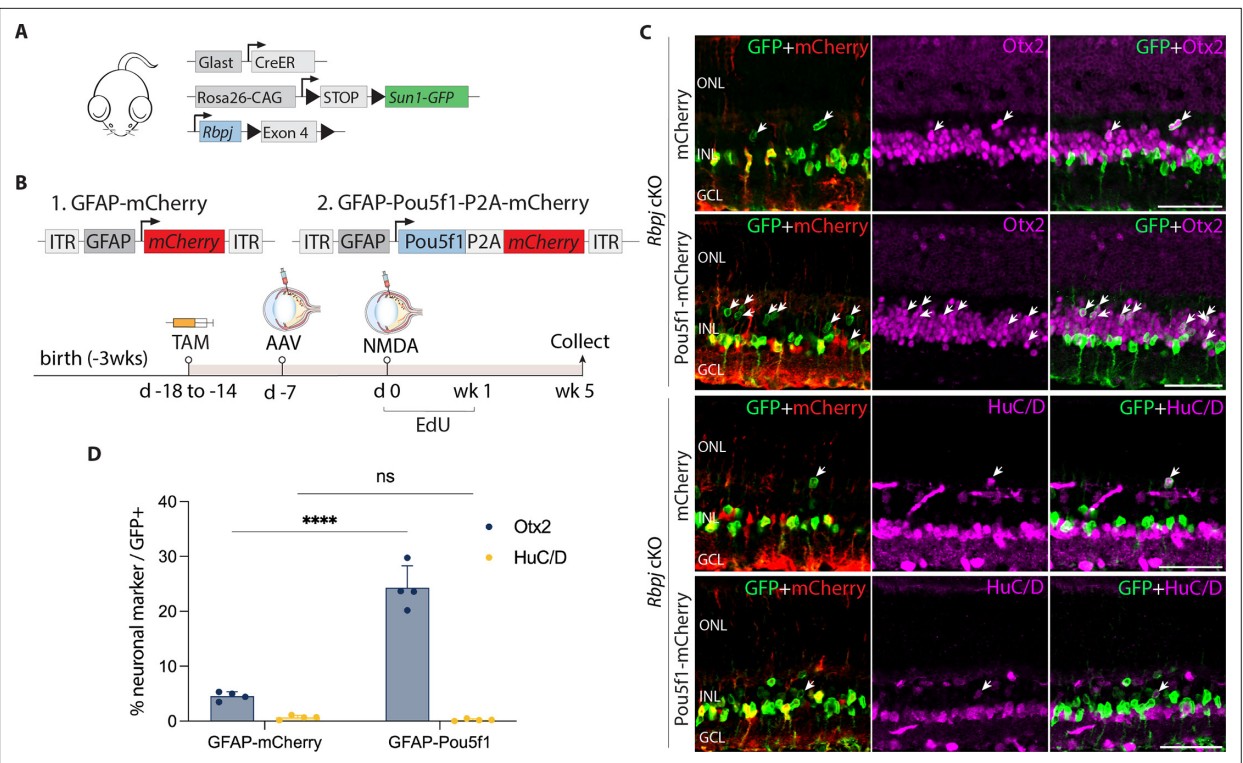

**Figure 2.** *Pou5f1* overexpression enhances neurogenesis in *Rbpj*-deficient GlastCreER;*Rbpj*^lox/lox^;*Rosa26*^LSL-Sun1-GFP^ Müller glia. (**A**) Schematic of the transgenic constructs used to induce loss of function of *Rbpj* specifically in Müller glia. (**B**) Schematic of the GFAP AAV constructs and experimental workflow. (**C**) Representative images of retinas immunolabeled for GFP, mCherry, Otx2, and HuC/D. White arrowheads indicate GFP-positive Müller glia-derived neurons expressing neuronal markers Otx2 or HuC/D. (**D**) Quantification of mean percentage ± SD of GFP-positive Müller glia-derived neurons expressing either Otx2 or HuC/D. Significance was determined via two-way ANOVA with Tukey's multiple comparison test: ****p < 0.0001. TAM, tamoxifen; ONL, outer nuclear layer; INL, inner nuclear layer; GCL, ganglion cell layer. Scale bar = 50 μm.

The online version of this article includes the following figure supplement(s) for figure 2:

**Figure supplement 1.** Immunohistochemical analysis of glial markers, AAV reporter expression, and other molecular markers at 1 and 2 weeks following *N*-methyl-D-aspartate (NMDA) in *Rbpj*-deficient Müller glia.

**Figure supplement 2.** Immunohistochemical analysis of glial markers, AAV reporter expression, and other molecular markers at 5 weeks following *N*-methyl-D-aspartate (NMDA) in *Rbpj*-deficient Müller glia.

**Figure supplement 3.** AAV-mediated overexpression of *Pou5f1* induces neurogenesis in injured *Notch1/2*-deficient Müller glia.

**Figure supplement 4.** *Pou5f1* overexpression does not enhance neurogenesis in *Nfia/b/x*-deficient Müller glia because the GFAP minipromoter is not active in these cells.

## *Pou5f1* overexpression synergistically enhances glial-derived neurogenesis induced by genetic disruption of Notch signaling

In our previous study, we demonstrated that disrupting Notch signaling in adult mouse Müller glia – either by selective deletion of the transcriptional mediator *Rbpj* or by combined disruption of *Notch1* and *Notch2* receptors – induces transdifferentiation into bipolar and amacrine-like cells (*Le et al., 2024*). To further determine the effects of *Pou5f1* overexpression in the absence of Notch signaling, we used GlastCreER;*Rbpj*^lox/lox^;*Rosa26*^LSL-Sun1-GFP^ mice to genetically disrupt *Rbpj* function (*Figure 2A*, *Figure 2—figure supplement 1a*). We then performed intravitreal injection of either control or *Pou5f1* overexpression constructs followed by NMDA and EdU injection as performed for GlastCreER;*Rosa26*^LSL-Sun1-GFP^ mice. Retinas were collected at 1, 2, and 5 weeks following NMDA treatment (*Figure 2B*, *Figure 2—figure supplement 1b*). Immunostaining confirmed expression of Pou5f1 protein in GFP-positive Müller glia of *Pou5f1-mCherry*-infected retinas, but not in age-matched *mCherry*-infected controls (*Figure 2—figure supplement 1c, d*, *Figure 2—figure supplement 2a*). We observed that many GFP-positive cells in *Pou5f1-mCherry*-infected retinas did not express the

Müller glial markers Sox2 (*Figure 2—figure supplement 1e, f*) or Sox9 (*Figure 2—figure supplement 2b*, yellow arrows), suggesting that they have lost their glial identity.

At 1 and 2 weeks post retinal injury, we observed selective expression of the neurogenic bHLH factor Ascl1 in both control mCherry and *Pou5f1-mCherry*-infected GFP-positive Rbpj-deficient Müller glia (*Figure 2—figure supplement 1g, h*). Time course analysis showed that Müller glia-derived cells expressing the bipolar cell marker Otx2 appeared as early as 1 week following NMDA injury (*Figure 2—figure supplement 1i–k*). However, we did not observe a significant change in the proportion of GFP-positive Rbpj-deficient Müller glia expressing the amacrine cell marker HuC/D. By 5 weeks after NMDA injury, we observed 4.5% of GFP-positive cells in retinas of mCherry-infected controls expressed Otx2, while 0.7% expressed HuC/D in *Pou5f1-mCherry*-infected retinas (*Figure 2C, D*). These levels of neurogenesis are consistent with previous findings in uninfected *Rbpj*-deficient Müller glia (*Le et al., 2024*). In contrast, we observed 24.3% of GFP-positive cells expressing Otx2 in *Pou5f1-mCherry*-infected retinas, representing a significant (p < 0.0001) increase relative to *mCherry* control. Unlike in control (GlastCreER;Rosa26$^{LSL-Sun1-GFP}$) retinas, *Pou5f1*-expressing, *Rbpj*-deficient Müller glia showed no EdU incorporation at either 1, 2, or 5 weeks following infection (*Figure 2—figure supplement 1l, m*, *Figure 2—figure supplement 2c*). EdU labeling instead colocalized with the microglia marker Iba1. The absence of Müller glia proliferation indicates that the observed neurogenesis is due largely to direct transdifferentiation of Müller glia into retinal neuron-like cells, rather than through dedifferentiation to a proliferative neuronal progenitor-like state.

We have previously shown that Müller glia-specific *Notch1/2* double loss of function mutants phenocopy neurogenesis seen in *Rbpj* mutants (*Le et al., 2024*). We then examined the effect of *Pou5f1* overexpression on GlastCreER;*Notch1*$^{lox/lox}$;*Notch2*$^{lox/lox}$;*Rosa26*$^{LSL-Sun1-GFP}$ mice using the same protocol previously described for control and *Rbpj*-deficient Müller glia (*Figure 2—figure supplement 3a, b*). We observed 15.8% of GFP+ Müller glia colocalized with Otx2 in *Pou5f1-mCherry*-infected retinas compared to 6.6% GFP/Otx2+ cells in samples infected with control mCherry vector (*Figure 2—figure supplement 3c, d*). As expected, infection with *Pou5f1-mCherry* vector induced both *Pou5f1* (*Figure 2—figure supplement 3e*) and Ascl1 (*Figure 2—figure supplement 3f*) expression in *Notch1/2*-deficient Müller glia.

Finally, we tested whether *Pou5f1* overexpression could enhance Müller glia-derived neurogenesis induced by loss of function of the transcription factors *Nfia*, *Nfib*, and *Nfix* (*Hoang et al., 2020*). To test this, we injected GlastCreER;*Nfia*$^{lox/lox}$;*Nfib*$^{lox/lox}$;*Nfix*$^{lox/lox}$;*Rosa26*$^{LSL-Sun1-GFP}$ mice with both *mCherry* and *Pou5f1-mCherry* vectors as previously described, then analyzed the retinas 5 weeks after NMDA injury (*Figure 2—figure supplement 4a, b*). We observed limited coexpression of mCherry and GFP following infection with either construct (*Figure 2—figure supplement 4c*), indicating that NFI factors may be required for appropriate glial-specific expression of the GFAP minipromoter construct, as suggested by previous findings in astrocytes (*Cebolla and Vallejo, 2006*).

## *Pou5f1* overexpression promotes expression of reactive glial markers and neurogenic bHLH factors in *Rbpj*-deficient Müller glia

To gain insight into the mechanism by which *Pou5f1* stimulates neurogenesis in *Rbpj*-deficient Müller glia, we conducted single-cell (sc)RNA-seq analysis of GFP-positive cells FACS-isolated from retinas of GlastCreER;*Rbpj*$^{lox/lox}$;*Rosa26*$^{LSL-Sun1-GFP}$ mice infected with either GFAP-*mCherry* control or GFAP-*Pou5f1-mCherry* virus. Mice were tamoxifen- and NMDA-treated as described in *Figure 2*, and GFP-positive cells were isolated at 8 weeks following NMDA treatment (*Figure 3A*). We profiled 12,143 GFP+ cells from control-infected and 14,612 GFP+ cells from *Pou5f1-mCherry*-infected retinas (*Figure 3B*).

In both samples, we observed a clear differentiation trajectory connecting Müller glia, neurogenic Müller glia-derived progenitor cells (MGPCs), and differentiating amacrine and bipolar cells (*Figure 3B*). *Pou5f1*-overexpressing samples consistently show substantially higher fractions of bipolar cells (20.5% vs. 2.7 %), and a smaller fraction of Müller glia (71.4% vs. 88.2 %), consistent with immunohistochemical data (*Figure 3C*). Small numbers of contaminating rod and cone photoreceptors were also observed, as previously reported (*Hoang et al., 2020*), but no immature photoreceptor-like cells were observed, indicating that these represent contaminating mature cells that were not excluded by FACS analysis.

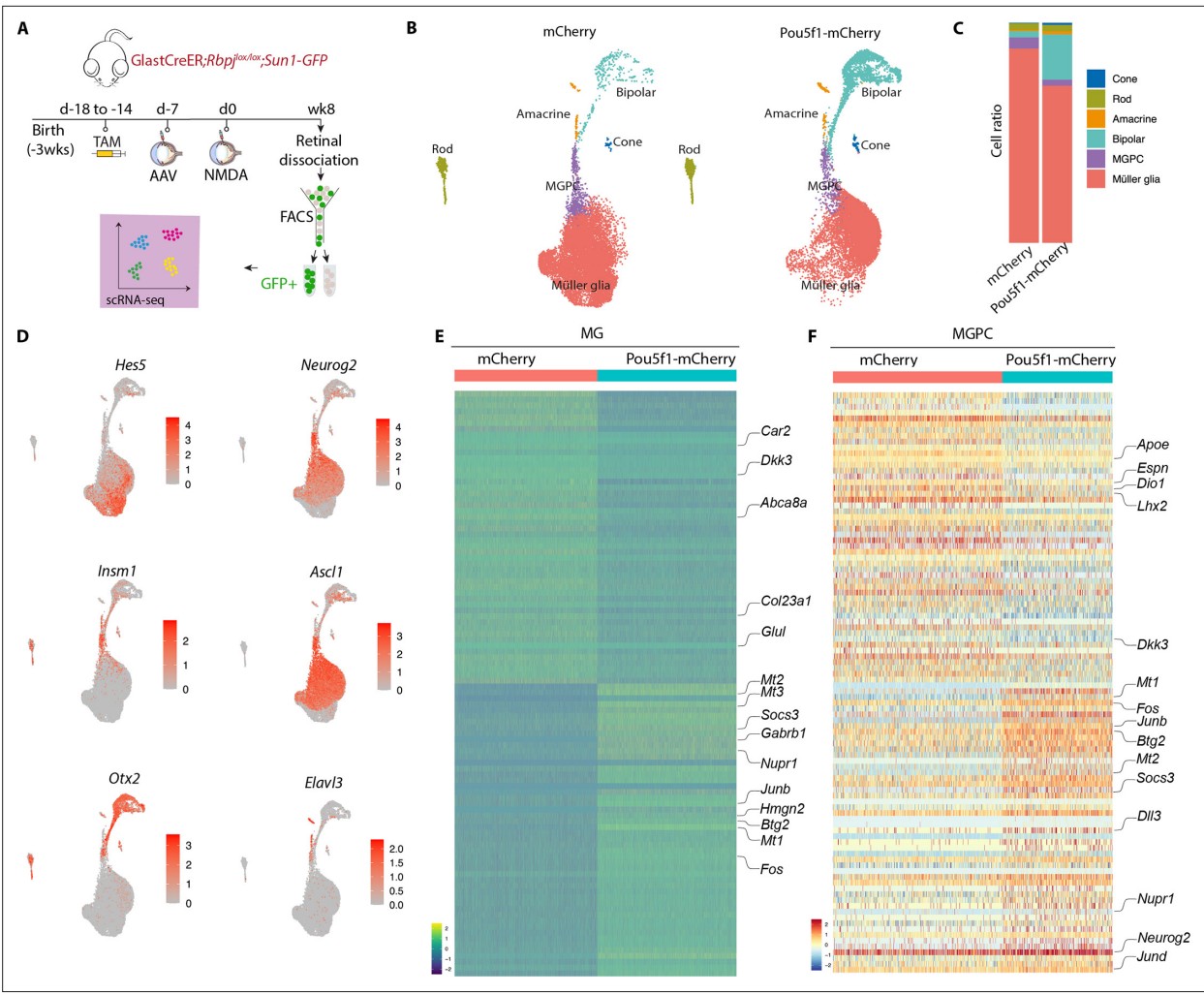

**Figure 3.** Single-cell RNA-sequencing (scRNA-seq) analysis of Müller glia and Müller glia-derived neurons from *Rbpj*-deficient GlastCreER;*Rbpj^lox/lox*;*Rosa26^LSL-Sun1-GFP* Müller glia following *Pou5f1* overexpression. (**A**) Schematic of the scRNA-seq experimental pipeline. (**B**) UMAP plot showing the clustering of GFP-positive cells from *Rbpj*-deficient GlastCreER;*Rbpj^lox/lox*;*Rosa26^LSL-Sun1-GFP* Müller glia retinas infected with GFAP-*mCherry* and GFAP-*Pou5f1-mCherry* AAV constructs. (**C**) Stacked bar plots showing the proportion of cells in each cluster across two sample groups. (**D**) Feature plots highlighting the cluster of Müller glia (*Hes5*), neurogenic Müller glia-derived progenitor cells (MGPCs) (*Neurog2*, *Insm1*, *Ascl1*), bipolar cells (*Otx2*), and amacrine cells (*Elavl3*). (**E**) Heatmap showing the expression of top differentially expressed genes (DEGs) from the *Rbpj*-deficient GlastCreER;*Rbpj^lox/lox*;*Rosa26^LSL-Sun1-GFP* Müller glia cell cluster from retinas infected with GFAP-*mCherry* and GFAP-*Pou5f1-mCherry* AAV constructs. (**F**) Heatmap showing the expression of top DEGs for MGPC cell cluster from retinas infected with GFAP-mCherry and GFAP-*Pou5f1-mCherry* AAV constructs.

The online version of this article includes the following figure supplement(s) for figure 3:

**Figure supplement 1.** Single-cell RNA-sequencing (scRNA-seq) analysis of Müller glia-derived neurons from *Rbpj*-deficient GlastCreER;*Rbpj^lox/lox*;*Rosa26^LSL-Sun1-GFP* Müller glia following *Pou5f1* overexpression.

Selective molecular markers could readily distinguish Müller glia (*Glul*, *Apoe*, *Vim*, *Clu*, *Rlbp1*) from neurogenic MGPCs (*Neurog2*, *Ascl1*, *Sox4*, *Hes6*, *Dll1*), and clearly identify differentiating bipolar (*Otx2*, *Pcp2*, *Pcp4*, *Car10*, *Gabrb3*) and amacrine-like cells (*Nrxn1*, *Tfap2b*, *Nrg1*, *Snap25*) (***Figure 3D–F***, ***Figure 3—figure supplement 1a–c***, ***Supplementary file 1***). In both MGs and MGPCs, we observed substantial differences in gene expression between samples infected with control and *Pou5f1-mCherry* samples. *Pou5f1*-infected Müller glia selectively downregulated genes specific to resting Müller glia (*Car2*, *Dkk3*, *Glul*, *Col23a1*, *Abca8a*), and upregulated genes enriched in reactive glia (*Gfap*, *Socs3*, *Nupr1*, *Mt1/2/3*, *Fos*, *Jun*, *Klf6*) (***Figure 3E, F***, ***Figure 3—figure supplement 1***, ***Supplementary file 1***). In MGPCs, *Pou5f1* overexpression also downregulated both transcription factors (*Lhx2*, *Hopx*) and other genes (*Apoe*, *Dio1*, *Il33*, *Espn*) that are enriched in resting Müller glia, and upregulated many other genes that are selectively expressed in activated glia (*Mt1/2/3*,

*Fos*, *Jun*, *Nupr1*, *Socs3*). In addition, *Pou5f1*-infected MGPCs expressed consistently higher levels of neurogenic bHLH factors (*Ascl1*, *Neurog2*, *Hes6*), bipolar cell specification (*Otx2*), or genes broadly associated with neurogenesis (*Insm1*, *Dll3*, *Mybl1*). *Pou5f1*-infected MGPCs also expressed genes (*Gadd45a*, *Btg2*) enriched in late-stage neurogenic progenitors (*Clark et al., 2019*; *Figure 3E, F*, *Supplementary file 1*). Together, these findings imply that *Pou5f1* enhances bipolar cell formation in *Rbpj*-deficient Müller glia potentially by repressing expression of genes such as *Lhx2* that promote quiescence (*de Melo et al., 2018*; *de Melo et al., 2016*), increasing expression of genes that broadly promote neurogenesis, and selectively increasing expression of the bipolar cell-promoting factor *Otx2* (*Chan et al., 2020*; *Wang et al., 2014*).

## Single-cell multiomic analysis identifies targets of *Pou5f1* in control and *Rbpj*-deficient Müller glia

To further investigate the mechanism by which *Pou5f1* promotes neurogenesis, we conducted simultaneous single-nucleus (sn)RNA-seq and ATAC-seq on GFP-positive cells from both control and *Rbpj*-deficient Müller glia infected with either control or *Pou5f1-mCherry* vectors at 8 weeks following NMDA injection (*Figure 4A*). UMAP analysis of integrated snRNA/ATAC-seq profiles was then used to identify cell clusters. Müller glia from control and *Rbpj*-deficient animals form two separate Müller glia clusters (*Figure 4B, C*). Control *Pou5f1*-expressing Müller glia gave rise to both small numbers of neurogenic MGPCs (*Figure 4D*), in line with the small numbers of Ascl1-positive cells observed (*Figure 1—figure supplement 2g, h*, *Figure 3—figure supplement 1b*), as well as mature bipolar cells, consistent with histological findings (*Figure 1C, D*). *Pou5f1* overexpression substantially enhances bipolar cell generation from *Rbpj*-deficient MGPC (*Figure 4D*, *Figure 4—figure supplement 1*). Small numbers of rod and cone photoreceptors were also detected in this analysis, but their proportions did not substantially change between treatment conditions and clearly defined immature transitional states were identified, and therefore likely represent contaminating native cells.

*Pou5f1* overexpression leads to significant changes in gene expression in both control and *Rbpj*-deficient Müller glia. The transition from resting to neurogenic MGPCs to bipolar cells is associated with downregulation of genes that are specifically expressed in resting Müller glia, including *Aqp4*, *Hes1*, *Tcf7l2*, and *Lhx2* (*Figure 4E, F*, *Figure 4—figure supplement 1a–c*, *Supplementary file 2*). Similarly, motif analysis showed that chromatin accessibility is reduced for target motifs for transcription factors that maintain glial quiescence such as Nfia/b/x and Lhx2 (*Figure 4G*, *Supplementary file 2*). MGPCs likewise show increased accessibility at Stat1/3 motifs, as well as target motifs for neurogenic transcription factors such as Ascl1, Neurog2, and Neurod1 (*Figure 4G*), consistent with previous studies of MGPCs generated from *Nfia/b/x* and *Rbpj*-deficient Müller glia (*Hoang et al., 2020*; *Le et al., 2024*).

*Pou5f1* overexpression in both control and *Rbpj*-deficient Müller glia clusters, respectively, induces upregulation of 534 and 1657 genes, while, respectively, downregulating 439 and 824 genes (*Figure 4H*). A total of 156 genes are upregulated in both control and *Rbpj*-deficient Müller glia, while 63 genes are downregulated. In both control and *Rbpj*-deficient Müller glia, *Pou5f1* upregulates multiple classes of ion channels, including mechanosensitive (*Piezo2*), TRP (*Trpm1*, *Trpm3*), and chloride channels (*Clca3a1*, *Clca3a2*), as well as GABA receptors (*Gabra4*, *Gabrb1*), cyclic nucleotide phosphodiesterases (*Pde8b*, *Pde11a*), and the neuropeptide *Npy*. *Pou5f1* overexpression also selectively downregulates multiple inflammation-associated genes (*B2m*, *H2-D1*), small GTPases (*Gbp2*, *Igtp*, *Irgm2*, *Rab13*, *Rgs20*, *Tgtp2*), along with the glycolytic enzyme *Gapdh* and many ribosomal subunits (*Figure 4H*, *Supplementary file 3*). This comprises a functionally diverse range of genes that are often not prominently expressed in retinal progenitors, and whose role in controlling neurogenic competence remains to be investigated.

*Pou5f1* also differentially regulates multiple transcription factors in both control and *Rbpj*-deficient Müller glia. As expected, *Pou5f1* mRNA is strongly upregulated (*Figure 4E, H*), with increased chromatin accessibility also being observed at the endogenous *Pou5f1* promoter, indicating that *Pou5f1* regulates its self-expression (*Figure 4I*, *Supplementary file 4*). Similarly, *Pou5f1* overexpression strongly upregulates the transcription factor *Rfx4* and accessibility at its endogenous promoter (*Figure 4E, H, I*, *Supplementary file 4*). Previous studies showed that overexpression of *Rfx4* is sufficient to induce neurogenesis when overexpressed in human ES cells (*Choi et al., 2024*; *Joung et al., 2024*). However, *Rfx4* expression is not detectably expressed in developing mouse retinas (*Clark*

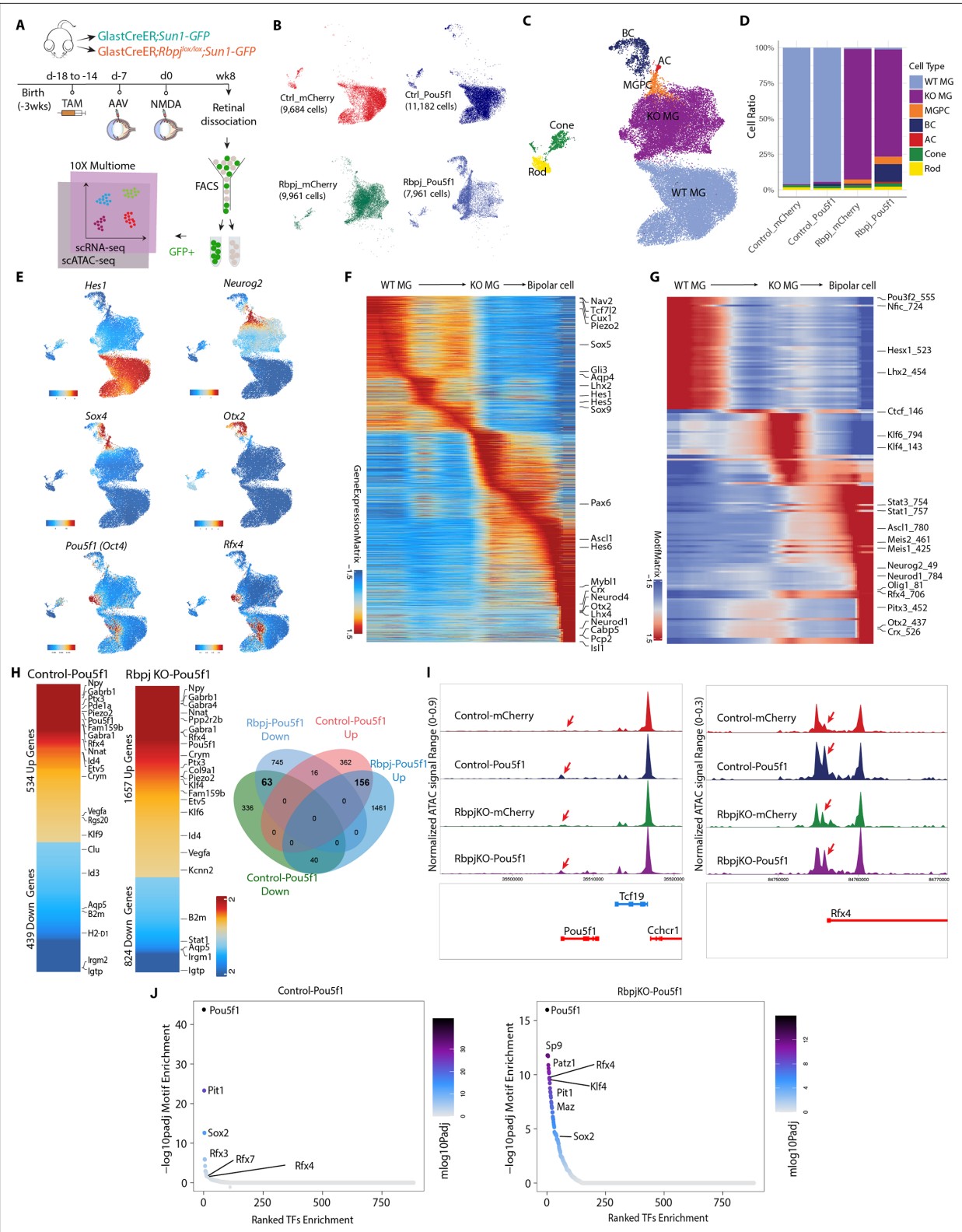

**Figure 4.** Integrated snRNA/scATAC-seq analysis of control GlastCreER;*Rosa26*^LSL-Sun1-GFP^ and *Rbpj*-deficient GlastCreER;*Rbpj*^lox/lox^;*Rosa26*^LSL-Sun1-GFP^ Müller glia following *Pou5f1* overexpression. (**A**) Schematic of the multiomic scRNA/ATAC-seq experimental pipeline. (**B**) UMAP plot of multiomic datasets showing the clustering of GFP+ cells from control GlastCreER;*Rosa26*^LSL-Sun1-GFP^ and *Rbpj*-deficient GlastCreER;*Rbpj*^lox/lox^;*Rosa26*^LSL-Sun1-GFP^ Müller glia from GFAP-*mCherry* and GFAP-*Pou5f1-mCherry* AAV infected retinas. (**C**) UMAP plot showing the identity of cell clusters determined by marker gene expression. (**D**) Stacked bar plots represent the proportion of cells in each cluster across different sample groups. (**E**) Feature plots highlighting the

*Figure 4 continued on next page*

*Figure 4 continued*

cluster of Müller glia (*Hes1*), neurogenic Müller glia-derived progenitor cells (MGPCs) (*Neurog2*, *Sox4*), bipolar cells (*Otx2*), *Pou5f1*- and *Rfx4*-expressing cells. (**F**) Heatmap showing expression of differentially expressed genes (DEGs) along the neurogenesis trajectory. (**G**) Heatmap showing differential motif activity along the neurogenesis trajectory. (**H**) Heatmaps of DEGs differentially expressed between *Pou5f1* and *mCherry* control samples in the control GlastCreER;*Rosa26*$^{LSL-Sun1-GFP}$ and *Rbpj*-deficient GlastCreER;*Rbpj*$^{lox/lox}$;*Rosa26*$^{LSL-Sun1-GFP}$ Müller glia clusters. Venn diagram showing unique and common DEGs between *Pou5f1* and *mCherry* control in the control GlastCreER;*Rosa26*$^{LSL-Sun1-GFP}$ and *Rbpj*-deficient GlastCreER;*Rbpj*$^{lox/lox}$;*Rosa26*$^{LSL-Sun1-GFP}$ Müller glia. (**I**) Increased chromatin accessibility regions associated with the *Pou5f1* and *Rfx4* loci observed following *Pou5f1* overexpression in both control GlastCreER;*Rosa26*$^{LSL-Sun1-GFP}$ and *Rbpj*-deficient GlastCreER;*Rbpj*$^{lox/lox}$;*Rosa26*$^{LSL-Sun1-GFP}$ Müller glia. (**J**) Top-ranked enriched motifs in chromatin regions showing increased accessibility following *Pou5f1* overexpression in both control GlastCreER;*Rosa26*$^{LSL-Sun1-GFP}$ and *Rbpj*-deficient GlastCreER;*Rbpj*$^{lox/lox}$;*Rosa26*$^{LSL-Sun1-GFP}$ Müller glia cell clusters.

The online version of this article includes the following figure supplement(s) for figure 4:

**Figure supplement 1.** Pseudotime analysis showing the transition from resting to neurogenic Müller glia-derived progenitor cells (MGPCs) to bipolar cells.

*et al., 2019*), indicating that *Pou5f1*-induced neurogenesis may involve mechanisms beyond simply inducing dedifferentiation to a retinal progenitor-like state.

Several other transcription factors are only differentially regulated following *Pou5f1* overexpression in *Rbpj*-deficient Müller glia. One notable upregulated transcription factor is *Klf4*, which cooperates with *Pou5f1* to induce pluripotency (*Takahashi and Yamanaka, 2006*). In contrast, *Stat1*, which inhibits neurogenesis induced by *Ascl1* overexpression in Müller glia (*Jorstad et al., 2020*), is selectively downregulated in *Rbpj*-deficient Müller glia (*Supplementary file 3*).

Analysis of changes in chromatin accessibility induced by *Pou5f1* reveals that the consensus motif for *Pou5f1* and its paralogue *Pit1* (*Pou1f1*) show the strongest increase in accessibility, followed by Rfx family motifs (*Figure 4J*, *Supplementary file 5*). Both control and *Rbpj*-deficient Müller glia show increased accessibility associated with Sox2, which directly interacts with *Pou5f1* and cooperates to induce pluripotency (*Rizzino, 2013*). While *Pou5f1* does not significantly upregulate the pluripotency factor *Klf4* in control Müller glia, it robustly induces *Klf4* expression *Rbpj*-deficient Müller glia, and likewise selectively induces accessibility at Klf4 consensus motif sites (*Figure 4J*). In addition, *Rbpj*-deficient Müller glia also show smaller but still significant increases in accessibility at target sites for the proneural factor Neurog2 and the neurogenic factor Insm1 (*Lyu et al., 2021*), identifying additional mechanisms by which *Pou5f1* might synergistically enhance neurogenesis (*Figure 4J*, *Supplementary file 5*).

## Discussion

In this study, we demonstrate that AAV-mediated overexpression of *Pou5f1* is sufficient to induce neurogenic competence in Müller glia in adult mice, and to synergistically enhance neurogenesis induced by genetic disruption of Notch signaling. While *Pou5f1* overexpression induces very low levels of proliferation, *Pou5f1*-induced neurogenesis appears to mostly result from direct glia-to-bipolar cell transdifferentiation by simultaneously upregulating expression of neurogenic bHLH factors and the bipolar cell-promoting factor *Otx2*. In addition, *Pou5f1* upregulates expression of the neurogenic transcription factor *Rfx4*, which is not expressed during retinal development. *Pou5f1* exerts reprogramming effects by broadly increasing chromatin accessibility at predicted regulatory sites associated with these genes.

While we did not detect endogenous *Pou5f1* expression in either injured or neurogenic mouse Müller glia, the fact that these cells express robust levels of *Sox2*, *Klf4*, and *Myc* implies that they may already be prime to reprogramming induced by *Pou5f1*. Our findings differ from two previous reports that initially motivated us to carry out this study. A previous study in zebrafish showed weak levels of the *Pou5f1* homolog *pou5f3* in resting glia, with these levels increasing dramatically following injury (*Sharma et al., 2019*). It remains unclear why we were unable to detect more than trace levels of *pou5f3* expression in zebrafish Müller glia in any of the many bulk and single-cell RNA-seq samples we analyzed (*Hoang et al., 2020*). We also did not detect the previously reported low levels of injury-induced *Pou5f1* expression in mouse Müller glia (*Reyes-Aguirre and Lamas, 2016*). Similar discrepancies in observed cellular patterns of *Pou5f1* expression have been observed in many other tissues (*Lengner et al., 2008*), and further work will be needed to clarify these discrepancies.

This study represents a clear instance in which viral-mediated expression coupled with genetic cell lineage analysis was effectively used to induce the conversion of adult mammalian Müller glia into retinal neurons in situ. A similar approach has been also successfully used to generate bipolar neurons from both immature (*Pollak et al., 2013*) and very recently also in mature, Müller glia by overexpression of the neurogenic bHLH factors *Ascl1* and *Atoh1* (*Pavlou et al., 2024*). Other studies reported efficient conversion of Müller glia to photoreceptors using a complex cocktail of overexpressed factors, including *Ctnnb1*, *Otx2*, *Crx*, and *Nrl* (*Yao et al., 2018*), and efficient conversion of Müller glia to retinal ganglion cells following overexpression of *Atoh7* and *Pou4f2* (*Xiao et al., 2021*). However, these studies both used GFAP minipromoters to both drive expression of these constructs and to infer cell lineage. Multiple studies have shown that GFAP minipromoter specificity is heavily influenced by insert sequences present in AAVs, which can result in both silencing of expression in glia and ectopic activation of expression in neurons (*Le et al., 2022*; *Wang et al., 2021b*). As a result, it cannot be concluded that these represent bona fide glia-to-neuron conversion in vivo. Similar claims have been made about knockdown of the RNA binding protein *Ptbp1*, which was variously claimed to directly convert Müller glia to either retinal ganglion cells or photoreceptors, but has been debunked using genetic cell lineage analysis (*Hoang et al., 2022*; *Xie et al., 2022*). In this study, the combination of prospective cell lineage analysis using the GlastCreER;*Rosa26^{LSL-Sun1-GFP}* transgene before infection with GFAP-*Pou5f1-mCherry* vector, in combination with single-cell RNA-sequencing (scRNA-seq) analysis of infected cells, rules out any potentially confounding factors resulting from the GFAP minipromoter.

Our finding that *Pou5f1* can synergistically enhance neurogenesis induced by genetic disruption of Notch signaling raises the question of whether this might be more broadly applicable to other models of induced neurogenesis in mammalian Müller glia. The mechanism by which *Pou5f1* induces glial-derived neurogenesis appears complex, and to differ fundamentally from that seen following either overexpression of *Ascl1* or loss of function of *Nfia/b/x* and/or *Rbpj* (*Hoang et al., 2020*; *Jorstad et al., 2017*; *Le et al., 2024*). In these cases, Müller glia broadly upregulate proneural genes and/ or downregulate Notch signaling. *Pou5f1* instead induces expression of the neurogenic transcription factor *Rfx4*, which is not expressed in developing retina. While this may represent a parallel pathway to neurogenic competence that in part accounts for synergistic induction of neurogenesis seen in *Rbpj*-deficient Müller glia, confirmation of the neurogenic activity of Rfx4 in Müller glia requires direct experimental confirmation.

Other mechanisms enhancing *Pou5f1*-dependent neurogenesis in the *Rbpj*-deficient Müller glia may include enhanced induction of *Klf4* expression and increased accessibility at target sites for neurogenic transcription factors such as Ascl1. While *Ascl1* overexpression induces Müller glia-derived neurogenesis in adult mice, this occurs only following treatment with HDAC inhibitors (*Jorstad et al., 2017*), even though Ascl1 is itself a pioneer factor (*Păun et al., 2023*). *Pou5f1* overexpression broadly increases the accessibility of putative cis-regulatory sequences associated with both *Neurog2* and *Insm1*, identifying another potential mechanism of synergy. Transient overexpression of *Pou5f1* could potentially further enhance the neurogenic function of Ascl1 and enhance glial-derived neurogenesis more broadly. This may also be the case for pathways reported to induce glial reprogramming in other CNS regions, such as *Sox2* overexpression (*Niu et al., 2015*). Since *Pou5f1* overexpression only weakly and transiently induces proliferation in adult Müller glia, this reduces the potential oncogenic risk, and may ultimately prove to be relevant for the design of cell-based regenerative therapies for treating retinal dystrophies.

## Methods

### Mice

Mice were raised and housed in a climate-controlled pathogen-free facility on a 14/10 hr light/dark cycle. Mice used in this study were GlastCreER;*Rosa26^{LSL-Sun1-GFP}*, which were generated by crossing the GlastCreER (JAX#012586) and *Rosa26^{LSL-Sun1-GFP}* (JAX#021039) lines developed by Dr. Jeremy Nathans at Johns Hopkins (*de Melo et al., 2012*; *Mo et al., 2015*), and were obtained from his group. Glast-CreER;*Rbpj^{lox/lox}*;*Rosa26^{LSL-Sun1-GFP}* mice were generated by crossing GlastCreER;*Sun1GFP* with conditional *Rbpj^{lox/lox}* mice (JAX #034200).

GlastCreER;*Notch1*<sup>lox/lox</sup>;*Notch2*<sup>lox/lox</sup>;*Rosa26*<sup>LSL-Sun1-GFP</sup> mice were obtained by crossing Glast-CreER;*Notch1*<sup>lox/lox</sup>;*Rosa26*<sup>LSL-Sun1-GFP</sup> and GlastCreER;*Notch2*<sup>lox/lox</sup>;*Rosa26*<sup>LSL-Sun1-GFP</sup> mice. GlastCreER;*Nfia/b/x*<sup>lox/lox</sup>;*Rosa26*<sup>LSL-Sun1-GFP</sup> mice were generated by crossing *Nfia*<sup>lox/lox</sup> (see below); *Nfib*<sup>lox/lox</sup> (**Hsu et al., 2011**), and *Nfix*<sup>lox/lox</sup> (**Campbell et al., 2008**) mice to GlastCreER;*Rosa26*<sup>LSL-Sun1-GFP</sup> mice. *Nfia*<sup>lox/lox</sup> mice were generated in the Roswell Park Gene Targeting and Transgenic Shared Resource using heterozygous targeted ES cells from EUCOMM project 38437 (KOMP). All mice used in this study contain both the GlastCreER and *Rosa26*<sup>LSL-Sun1-GFP</sup> transgenes, allowing visualization of both Müller glia and Müller glia-derived neurons. Maintenance and experimental procedures performed on mice were in accordance under protocol MO22M22 approved by the Institutional Animal Care and Use Committee (IACUC) at the Johns Hopkins School of Medicine.

## Intraperitoneal tamoxifen injection

To induce *Cre* recombination, animals at ~3 weeks of age were intraperitoneally injected with tamoxifen (Sigma-Aldrich, #H6278-50mg) in corn oil (Sigma-Aldrich, #C8267-500ML) at 1.5 mg/dose for 5 consecutive days.

## NMDA treatment

Adult mice were anesthetized with isoflurane inhalation. Two microliters of 100 mM NMDA in PBS were intravitreally injected using a microsyringe with a 33 G blunt-ended needle.

## EdU treatment

At 1 week post tamoxifen induction, mice were administered EdU (Abcam, #ab146186) via i.p. injections (150 µl of 5 mg/ml in PBS) and drinking water (0.3 mg/ml) for 7 consecutive days.

## Cloning, production, and intravitreal injection of adeno-associated virus

The Addgene #50473 construct, which contains a GFAP promoter, was used in this study. The EGFP sequence was replaced by the *mCherry* sequence. The P2A ribosomal self-cleaving peptide is used to simultaneously express *Pou5f1* 5′ to the *mCherry* reporter as a single polypeptide, which is then cleaved to generate *Pou5f1* and *mCherry*. The coding sequence of *Pou5f1* was synthesized by GeneWiz. AAV constructs were packaged into AAV2.7m8 by Boston Children's Hospital Viral Core. Following tamoxifen induction, 1-month-old GlastCreER;*Rosa26*<sup>LSL-Sun1-GFP</sup> mice were intravitreally injected with GFAP AAV constructs using a microsyringe with a 33 G blunt-ended needle. The titers and injection volume for each construct are listed in **Supplementary file 6**.

## Immunohistochemistry and imaging

Collection and immunohistochemical analysis of retinas were performed as described previously (**Hoang et al., 2020**). Mouse eye globes were fixed in 4% paraformaldehyde (ElectronMicroscopySciences, #15710) for 4 hr at 4°C. Retinas were dissected in 1× PBS and incubated in 30% sucrose overnight at 4°C. Retinas were then embedded in OCT (VWR, #95057-838), cryosectioned at 16 µm thickness, and stored at −20°C. Sections were dried for 30 min in a 37°C incubator and washed 3 × 5 min with 0.1% Triton X-100 in PBS (PBST). EdU labeling was performed by using Click-iT EdU kit (Thermo Fisher, #C10340, #C10636) following the manufacturer's instructions. Sections were then incubated in blocking buffer (10% Horse Serum (Thermo Fisher, #26050070), 0.4% Triton X-100 in 1× PBS) for 2 hr at room temperature (RT) and then incubated with primary antibodies in the blocking buffer overnight at 4°C. Primary antibodies used are listed in Appendix 1—key resources table.

Sections were washed 4 × 5 min with PBST to remove excess primary antibodies and were incubated in secondary antibodies in blocking buffer for 2 hr at RT. Sections were then counterstained with DAPI in PBST, washed 4 × 5 min in PBST and mounted with ProLong Gold Antifade Mountant (Invitrogen, #P36935) under coverslips (VWR, #48404-453), air-dried, and stored at 4°C. Fluorescent images were captured using a Zeiss LSM 700 confocal microscope. Z-stack images were collected for all sections. Colocalization of markers scored only if observed in individual Z-stack images. Secondary antibodies used are listed in Appendix 1—key resources table.

## Cell quantification and statistical analysis

Otx2/GFP- and HuC/D/GFP-positive cells were counted and divided by the total number of GFP-positive cells from a single random whole section per retina. Each data point in the bar graphs was

calculated from an individual retina. All cell quantification data were graphed and analyzed using GraphPad Prism 10. Two-way ANOVA was used for analysis between three or more samples of multiple groups. All results are presented as mean ± SD.

## Retinal cell dissociation

Retinas were dissected in fresh ice-cold PBS and retinal cells were dissociated using an optimized protocol as previously described (*Fadl et al., 2020*). Each sample contains a minimum of four retinas from four animals of both sexes. Dissociated cells were resuspended in ice-cold HBAG Buffer containing Hibernate A (BrainBits, #HALF500), B-27 supplement (Thermo Fisher, #17504044), and Glutamax (Thermo Fisher, #35050061).

## scRNA-seq library preparation

ScRNA-seq libraries were prepared using dissociated retinal cells using the 10X Genomics Chromium Single Cell 3′ Reagents Kit v3.1 (10X Genomics, Pleasanton, CA). Libraries were constructed following the manufacturer's instructions and were sequenced using Illumina NextSeq. Sequencing data were processed through the Cell Ranger 7.0.1 pipeline (10X Genomics) using default parameters.

## Single-cell Multiome ATAC + GEX sequencing library preparation

ScATAC-seq and scRNA-seq libraries were prepared using FACS-isolated GFP-positive cells using the 10X Genomic Chromium Next GEM Single Cell Multiome ATAC + Gene Expression kit following the manufacturer's instructions. Briefly, cells were spun down at $500 \times g$ for 5 min, resuspended in 100 µl of ice-cold 0.1× Lysis Buffer, lysed by pipette-mixing four times, and incubated on ice for 4 min total. Cells were washed with 0.5 ml of ice-cold Wash Buffer and spun down at $500 \times g$ for 5 min at 4°C. Nuclei pellets were resuspended in 10–15 µl Nuclei Buffer and counted using Trypan blue. Resuspended cell nuclei (10–15k) were utilized for transposition and loaded into the 10 Genomics Chromium Single Cell system. ATAC libraries were amplified with 10 PCR cycles and were sequenced on Illumina NovaSeq with ~200 million reads per library. RNA libraries were amplified from cDNA with 14 PCR cycles and were sequenced on Illumina NovaSeq 6000.

## scRNA-seq data analysis

ScRNA-seq data were pre-processed using Cellranger v7.1.0 using a custom mouse genome (version mm10) with mCherry and Sun1GFP sequences included. The cell-by-genes count matrices were further analyzed using Seurat V5.1.0 (*Hao et al., 2024*). Cells with RNA counts less than 1000 or greater than 25,000, and number of genes less than 500 or greater than 6000 were filtered out as low-quality cells. Cells with a mitochondrial fraction of greater than 10% were removed. For data visualization, UMAP was generated using the first 20 dimensions. Müller glia and MGPC cell clusters were subsetted for further analysis. Differential gene expression of *mCherry* control and *Pou5f1-mCherry* groups was performed using the 'FindAllMarkers' function.

## Single-cell multiomic analysis

Raw scRNA-seq and scATAC-seq data were processed with the Cell Ranger software version 2.0.2 specifically using cellranger-arc pipeline for formatting reads, demultiplexing samples, genomic alignment, and generating the cell-by-gene count matrix and the fragments files. Both the cell-by-gene count matrix and the fragments files were the final output from the cellranger-arc pipeline, and were used for all downstream analysis.

ScRNA-seq and scATAC-seq were analyzed using Archr version 1.0.2. EnsDb.Mmusculus.v79 and BSgenome.Mmusculus.UCSC.mm10 were used for annotations. The arrow file for each sample was created using the cell-by-gene count matrix and the fragments files output from the cellranger-arc pipeline. To filter poor quality cells, Gex_nGenes greater than 500 and less than 5000 were kept, Gex_nUMI greater than 1000 and less than 15,000 were kept. TSSEnrichment greater than 10 and nFrags greater than 5000 were kept, while other cells were deemed of poor quality and were filtered out. Gene expression values from paired scATAC-seq and scRNA-seq multi modal assay were added to the arrow files using the 'addGeneExpressionMatrix' function. The LSI dimensionality reduction for both scATAC and scRNASeq was computed and combined using the 'addCombinedDims' function. UMAP

embedding is calculated using 'addUMAP' function and clusters are identified from the reduced dimensions. Marker features were used to annotate clusters.

To identify peaks, 'addReproduciblePeakSet' function was used to get insertions from coverage files, call peaks, and merge peaks and this step internally used macs software. Then the 'addPeak-Matrix' function was called to independently compute counts for each peak per cell. To link peaks to genes, 'getPeak2GeneLinks' function was called. The 'getMarkerFeatures' function was used to get the differential gene expressions and the differential peaks expressions using the GeneExpressionMatrix and PeakMatrix calculated, respectively. Peaks with motifs were annotated using the cisbp motif set using 'annotatePeaks.pl' from HOMMER version 4.11 to associate the peaks with nearby genes. The full code used for the analysis is available at https://github.com/SherineAwad/OCT4scRNA-ATAC, copy archived at *Awad, 2025*.

## Acknowledgements

This study was supported by an award from the National Eye Institute (R01EY031685) to SB, a Stein Innovation Award from Research to Prevent Blindness to SB, and a Young Investigator Award from Alcon Research Institute to TH.

## Additional information

### Competing interests
Seth Blackshaw: is a co-founder, Scientific Director, and shareholder of CDI Labs, LLC. He also receives support from Genentech. The other authors declare that no competing interests exist.

### Funding

| Funder | Grant reference number | Author |
|---|---|---|
| National Eye Institute | R01EY031685 | Seth Blackshaw |
| Research to Prevent Blindness | Stein Innovation Award | Seth Blackshaw |
| Alcon Research Institute | Young Investigator Award | Thanh Hoang |

The funders had no role in study design, data collection, and interpretation, or the decision to submit the work for publication.

### Author contributions
Nguyet Le, Conceptualization, Data curation, Formal analysis, Validation, Investigation, Methodology, Writing – original draft, Writing – review and editing; Sherine Awad, Data curation, Formal analysis, Methodology, Writing – review and editing; Isabella Palazzo, Data curation, Formal analysis, Investigation, Writing – review and editing; Thanh Hoang, Conceptualization, Formal analysis, Supervision, Funding acquisition, Investigation, Writing – original draft, Project administration, Writing – review and editing; Seth Blackshaw, Conceptualization, Supervision, Funding acquisition, Methodology, Writing – original draft, Project administration, Writing – review and editing

### Author ORCIDs

Sherine Awad ⬛ https://orcid.org/0000-0002-8606-0674
Seth Blackshaw ⬛ https://orcid.org/0000-0002-1338-8476

### Ethics
Maintenance and experimental procedures performed on mice were in accordance under protocol MO22M22 approved by the Institutional Animal Care and Use Committee (IACUC) at the Johns Hopkins School of Medicine.

Reviewer #1 (Public review): https://doi.org/10.7554/eLife.106450.3.sa1
Reviewer #2 (Public review): https://doi.org/10.7554/eLife.106450.3.sa2

Author response https://doi.org/10.7554/eLife.106450.3.sa3

## Additional files

### Supplementary files

Supplementary file 1. ScRNA-seq analysis of Sun1-GFP Müller glia-derived cells identifies genes differentially regulated by *Pou5f1* overexpression. Raw + adjusted p-values, relative fold-change, and the fraction of all profiled cells expressing the gene in question is shown for both Muller glia and Muller glia-derived progenitor cells transfected with AAV-GFAP-*Pou5f1-mCherry* vs. AAV-GFAP-*mCherry*.

Supplementary file 2. Multiomic analysis identifies differentially expressed genes (DEGs), differential chromatin accessible regions, and differentially accessible transcription factor target motifs are shown at discrete pseudotime intervals along the Müller glia-bipolar differentiation trajectory.

Supplementary file 3. SnRNA-seq analysis identifies differentially expressed genes (DEGs) induced by *Pou5f1* overexpression from control GlastCreER;*Rosa26*$^{LSL-Sun1-GFP}$ and *Rbpj*-deficient GlastCreER;*Rbpj*$^{lox/lox}$;*Rosa26*$^{LSL-Sun1-GFP}$ Müller glia cell clusters.

Supplementary file 4. SnATAC-seq analysis identifies differentially accessible chromatin regions regulated by *Pou5f1* overexpression in control GlastCreER;*Rosa26*$^{LSL-Sun1-GFP}$ and *Rbpj*-deficient GlastCreER;*Rbpj*$^{lox/lox}$;*Rosa26*$^{LSL-Sun1-GFP}$ Müller glia cell clusters.

Supplementary file 5. SnATAC-seq analysis identifies differential motif activity in increased accessible chromatin regions by *Pou5f1* overexpression in control GlastCreER;*Rosa26*$^{LSL-Sun1-GFP}$ and *Rbpj*-deficient GlastCreER;*Rbpj*$^{lox/lox}$;*Rosa26*$^{LSL-Sun1-GFP}$ Müller glia cell clusters.

Supplementary file 6. List of AAV constructs used in the study. Columns identify the AAV construct, titer, and injection volume.

MDAR checklist

### Data availability

All scRNA-seq, snRNA-seq, and scATAC-seq data described in this study are available at Gene Expression Omnibus under accession number GSE277390.

The following dataset was generated:

| Author(s) | Year | Dataset title | Dataset URL | Database and Identifier |
|---|---|---|---|---|
| Le N, Awad S, Palazzo I, Hoang T, Blackshaw S | 2024 | Oct4 overexpression and suppression of Notch signaling synergistically induce neurogenic competence in mammalian Muller glia | https://www.ncbi.nlm.nih.gov/geo/query/acc.cgi?acc=GSE277390 | NCBI Gene Expression Omnibus, GSE277390 |

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

# Appendix 1

## Appendix 1—key resources table

| Reagent type (species) or resource | Designation | Source or reference | Identifiers | Additional information |
|---|---|---|---|---|
| Antibody | Chicken anti-GFP (chicken polyclonal) | ThermoFisher | A10262, RRID:AB_2534023 | IF (1:400) |
| Antibody | Rabbit anti-RFP (rabbit polyclonal) | Abcam | ab124754, RRID:AB_10971665 | IF (1:400) |
| Antibody | Goat anti-RFP (goat polyclonal) | Rockland | 200-101-379, RRID:AB_2744552 | IF (1:400) |
| Antibody | Goat anti-Otx2 (goat polyclonal) | R&D Systems | AF1979, RRID:AB_2157172 | IF (1:200) |
| Antibody | Mouse anti-HuC/D (mouse monoclonal) | ThermoFisher | A-21271, RRID:AB_221448 | IF (1:200) |
| Antibody | Mouse anti-NeuN (mouse monoclonal) | Sigma-Aldrich | MAB377, RRID:AB_2298772 | IF (1:200) |
| Antibody | Mouse anti-Oct3/4 (mouse monoclonal) | Santa Cruz | sc-5279, RRID:AB_628051 | IF (1:200) |
| Antibody | Rabbit anti-Sox9 (mouse monoclonal) | Sigma-Aldrich | AB5535, RRID:AB_2239761 | IF (1:400) |
| Antibody | Goat anti-Nrl (goat polyclonal) | R&D Systems | AF2945, RRID:AB_2155098 | IF (1:400) |
| Antibody | Rabbit anti-Scgn (rabbit polyclonal) | Biovendor Laboratory Medicine | RD181120100, RRID:AB_2034060 | IF (1:400) |
| Antibody | Rabbit anti-Ascl1 (rabbit monoclonal) | Abcam | ab211327, RRID:AB_2924270 | IF (1:400) |
| Antibody | Rabbit anti-CaBP5 (rabbit monoclonal) | SynapticSystems | 475 002, RRID:AB_2924962 | IF (1:400) |
| Antibody | Goat anti-Sox2 (goat polyclonal) | R&D Systems | AF2018, RRID:AB_355110 | IF (1:400) |
| Antibody | Mouse anti-Isl1 (mouse monoclonal) | DSHB | 40.2D6, RRID:AB_528315 | IF (1:400) |
| Antibody | Donkey anti-Chicken 488 (donkey polyclonal) | Sigma-Aldrich | SAB4600031, RRID:AB_2721061 | IF (1:400) |
| Antibody | Donkey anti-Rabbit 568 (donkey polyclonal) | ThermoFisher | A-10042, RRID:AB_2757564 | IF (1:400) |
| Antibody | Donkey anti-Goat 568 (donkey polyclonal) | ThermoFisher | A11057, RRID:AB_2534104 | IF (1:400) |
| Antibody | Donkey anti-Mouse 568 (donkey polyclonal) | ThermoFisher | A10037, RRID:AB_2534013 | IF (1:400) |
| Antibody | Donkey anti-Rat 568 (donkey polyclonal) | ThermoFisher | A78946, RRID:AB_2910653 | IF (1:400) |
| Antibody | Donkey anti-Goat 633 (donkey polyclonal) | ThermoFisher | A-21082, RRID:AB_2535739 | IF (1:400) |
| Antibody | Donkey anti-Rabbit 647 (donkey polyclonal) | ThermoFisher | A-31573, RRID:AB_2536183 | IF (1:400) |
| Antibody | Donkey anti-Mouse 647 (donkey polyclonal) | ThermoFisher | A-31571, RRID:AB_162542 | IF (1:400) |
| Commercial assay or kit | 10 x scRNaseq 3' v3.1 | 10 X Genomics | 1000268 | |
| Commercial assay or kit | 10 x Multiome ATAC +GEX | 10 X Genomics | 1000283 | |
| Commercial assay or kit | Click-iT EdU Alexa Fluor 647 | ThermoFisher | C10340 | |

*Appendix 1 Continued on next page*

*Appendix 1 Continued*

| Reagent type (species) or resource | Designation | Source or reference | Identifiers | Additional information |
|---|---|---|---|---|
| Commercial assay or kit | Click-iT EdU Pacific Blue | ThermoFisher | C10418 | |
| Software, algorithm | ImageJ/Fiji | https://imagej.net/software/fiji/ | RRID:SCR_002285 | |
| Software, algorithm | Adobe Illustrator | http://www.adobe.com | RRID:SCR_010279 | v.26.5 |
| Software, algorithm | GraphPad Prism | https://www.graphpad.com/ | RRID:SCR_002798 | v.10 |
| Software, algorithm | Cell Ranger | 10 X Genomics | RRID:SCR_017344 | v.2.0.2 |
| Software, algorithm | ArchR | https://github.com/GreenleafLab/ArchR | RRID:SCR_020982 | v.1.0.2 |
| Software, algorithm | Seurat | https://github.com/satijalab/seurat | RRID:SCR_007322 | v.5.1.0 |

