## [Editor Report · eLife Assessment]

This manuscript demonstrates that Oct4 overexpression synergizes with Notch inhibition (Rbpj knockout) to promote the conversion of adult murine Müller glia (MG) into bipolar cells. These findings are **important** as the authors used rigorous genetic lineage tracing (GLAST-CreER; Sun-GFP) to confirm that neurogenesis indeed originates from MGs, addressing a key issue in the field. The single-cell multiomic analyses are **compelling**, and while functional studies of MG-derived bipolar cells would strengthen the conclusions, they are beyond the scope of this study.

---

## [Referee Report · Reviewer #1 (Public review)]

Summary:

In this study, Le et al.. aimed to explore whether AAV-mediated overexpression of Oct4 could induce neurogenic competence in adult murine Müller glia, a cell type that, unlike its counterparts in cold-blooded vertebrates, lacks regenerative potential in mammals. The primary goal was to determine whether Oct4 alone, or in combination with Notch signaling inhibition, could drive Müller glia to transdifferentiate into bipolar neurons, offering a potential strategy for retinal regeneration.

The authors demonstrated that Oct4 overexpression alone resulted in the conversion of 5.1% of Müller glia into Otx2+ bipolar-like neurons by five weeks post-injury, compared to 1.1% at two weeks. To further enhance the efficiency of this conversion, they investigated the synergistic effect of Notch signaling inhibition by genetically disrupting Rbpj, a key Notch effector. Under these conditions, the percentage of Müller glia-derived bipolar cells increased significantly to 24.3%, compared to 4.5% in Rbpj-deficient controls without Oct4 overexpression. Similarly, in Notch1/2 double-knockout Müller glia, Oct4 overexpression increased the proportion of GFP+ bipolar cells from 6.6% to 15.8%.

To elucidate the molecular mechanisms driving this reprogramming, the authors performed single-cell RNA sequencing (scRNA-seq) and ATAC-seq, revealing that Oct4 overexpression significantly altered gene regulatory networks. They identified Rfx4, Sox2, and Klf4 as potential mediators of Oct4-induced neurogenic competence, suggesting that Oct4 cooperates with endogenously expressed neurogenic factors to reshape Müller glia identity.

Overall, this study aimed to establish Oct4 overexpression as a novel and efficient strategy to reprogram mammalian Müller glia into retinal neurons, demonstrating both its independent and synergistic effects with Notch pathway inhibition. The findings have important implications for regenerative therapies as they suggest that manipulating pluripotency factors in vivo could unlock the neurogenic potential of Müller glia for treating retinal degenerative diseases.

Strengths:

(1) Novelty: The study provides compelling evidence that Oct4 overexpression alone can induce Müller glia-to-bipolar neuron conversion, challenging the conventional view that mammalian Müller glia lacks neurogenic potential.

(2) Technological Advances: The combination of Muller glia-specific labeling and modifying mouse line, AAV-GFAP promoter-mediated gene expression, single-cell RNA-seq, and ATAC-seq provides a comprehensive mechanistic dissection of glial reprogramming.

(3) Synergistic Effects: The finding that Oct4 overexpression enhances neurogenesis in the absence of Notch signaling introduces a new avenue for retinal repair strategies.

Weaknesses:

(1) In this study, the authors did not perform a comprehensive functional assessment of the bipolar cells derived from Müller glia to confirm their neuronal identity and functionality.

(2) Demonstrating visual recovery in a bipolar cell-deficiency disease model would significantly enhance the translational impact of this work and further validate its therapeutic potential.

Comments on revisions:

The author answered all my questions and corrected the minor comments, so I have no more comments on the manuscript.

---

## [Referee Report · Reviewer #2 (Public review)]

Summary:

The authors harness single cell RNAseq data from zebrafish and mice to identify Oct4 as a candidate driver of neurogenesis. They then use adeno-associated virus vectors to show that while Oct4 overexpression alone converts rare adult Müller glia (MG) to bipolar cells, it synergizes with Notch pathway inhibition to cause this neurogenesis (achieved by Cre-mediated knockout of Rbpj floxed allele). Importantly, they genetically lineage-mark adult MG using a GLAST-CreER transgene and a Sun-GFP reporter, so that any non-MG cells that convert can be identified unambiguously. This is crucial because several high-profile papers made erroneous claims using short promoters in the viral delivery vector itself to mark MG, but those promoters are leaky and mark other non-MG cell types, making it impossible to definitively state whether manipulations studied were actually causing neurogenesis, or were merely the result of expression in pre-existing neurons. Once the authors establish Oct4 + RbpjKO synergy they use snRNAseq/ATACseq to identify known and novel transcription factors that could play a role in driving neurogenesis.

Strengths:

The system to mark MG is stringent, so the authors are studying transdifferentiation, not artifactual effects due to leaky viral promoters. The synergy between Oct4 and Notch pathway blockade is notable. The single cell results add the potential involvement of new players such as Rfx4 in adult-MG-neurogenesis.

Weaknesses:

The revised version is clear and there are no major weaknesses.

Overall, the authors achieved what they set out to do, and have made new insights into how neurogenesis can be stimulated in MG. Ultimately, a major long-term goal in the field is to replace lost photoreceptors as this is most relevant to many human visual disorders, and while this paper (like all others before it) does not generate rods or cones, it opens new strategies to coax MG to form a related neuronal cell type. Their approach underscores the benefits of using a gold standard approach for lineage tracing.

---

## [Author Response]

The following is the authors’ response to the original reviews

**Public Reviews:**

**Reviewer #1 (Public review):**
Summary:In this study, Le et al.. aimed to explore whether AAV-mediated overexpression of Oct4 could induce neurogenic competence in adult murine Müller glia, a cell type that, unlike its counterparts in cold-blooded vertebrates, lacks regenerative potential in mammals. The primary goal was to determine whether Oct4 alone, or in combination with Notch signaling inhibition, could drive Müller glia to transdifferentiate into bipolar neurons, offering a potential strategy for retinal regeneration.The authors demonstrated that Oct4 overexpression alone resulted in the conversion of 5.1% of Müller glia into Otx2+ bipolar-like neurons by five weeks post-injury, compared to 1.1% at two weeks. To further enhance the efficiency of this conversion, they investigated the synergistic effect of Notch signaling inhibition by genetically disrupting Rbpj, a key Notch effector. Under these conditions, the percentage of Müller gliaderived bipolar cells increased significantly to 24.3%, compared to 4.5% in Rbpjdeficient controls without Oct4 overexpression. Similarly, in Notch1/2 double-knockout Müller glia, Oct4 overexpression increased the proportion of GFP+ bipolar cells from 6.6% to 15.8%.To elucidate the molecular mechanisms driving this reprogramming, the authors performed single-cell RNA sequencing (scRNA-seq) and ATAC-seq, revealing that Oct4 overexpression significantly altered gene regulatory networks. They identified Rfx4, Sox2, and Klf4 as potential mediators of Oct4-induced neurogenic competence, suggesting that Oct4 cooperates with endogenously expressed neurogenic factors to reshape Müller glia identity.Overall, this study aimed to establish Oct4 overexpression as a novel and efficient strategy to reprogram mammalian Müller glia into retinal neurons, demonstrating both its independent and synergistic effects with Notch pathway inhibition. The findings have important implications for regenerative therapies as they suggest that manipulating pluripotency factors in vivo could unlock the neurogenic potential of Müller glia for treating retinal degenerative diseases.Strengths:(1) Novelty: The study provides compelling evidence that Oct4 overexpression alone can induce Müller glia-to-bipolar neuron conversion, challenging the conventional view that mammalian Müller glia lacks neurogenic potential.(2) Technological Advances: The combination of Muller glia-specific labeling and modifying mouse line, AAV-GFAP promoter-mediated gene expression, single-cell RNA-seq, and ATAC-seq provides a comprehensive mechanistic dissection of glial reprogramming.(3) Synergistic Effects: The finding that Oct4 overexpression enhances neurogenesis in the absence of Notch signaling introduces a new avenue for retinal repair strategies.Weaknesses:(1) In this study, the authors did not perform a comprehensive functional assessment of the bipolar cells derived from Müller glia to confirm their neuronal identity and functionality.(2) Demonstrating visual recovery in a bipolar cell-deficiency disease model would significantly enhance the translational impact of this work and further validate its therapeutic potential.

Response: We thank the Reviewer for their evaluation. We agree that functional analysis of Müller glia-derived bipolar cells is indeed important, but is beyond the current scope of the manuscript.

**Reviewer #2 (Public review):**
Summary:The authors harness single-cell RNAseq data from zebrafish and mice to identify Oct4 as a candidate driver of neurogenesis. They then use adeno-associated virus vectors to show that while Oct4 overexpression alone converts rare adult Müller glia (MG) to bipolar cells, it synergizes with Notch pathway inhibition to cause this neurogenesis (achieved by Cre-mediated knockout of Rbpj floxed allele). Importantly, they genetically lineage-mark adult MG using a GLAST-CreER transgene and a Sun-GFP reporter, so that any non-MG cells that convert can be identified unambiguously. This is crucial because several high-profile papers made erroneous claims using short promoters in the viral delivery vector itself to mark MG, but those promoters are leaky and mark other non-MG cell types, making it impossible to definitively state whether manipulations studied were actually causing neurogenesis, or were merely the result of expression in pre-existing neurons. Once the authors establish Oct4 + RbpjKO synergy they use snRNAseq/ATACseq to identify known and novel transcription factors that could play a role in driving neurogenesis.Strengths:The system to mark MG is stringent, so the authors are studying transdifferentiation, not artifactual effects due to leaky viral promoters. The synergy between Oct4 and Notch pathway blockade is notable. The single-cell results add the potential involvement of new players such as Rfx4 in adult-MG-neurogenesis.Weaknesses:The existing version is difficult to read due to an unusually high number of text errors (e.g. references to the wrong figure panels etc.). A fuller explanation for the fraction of non-MG cells seen in control scRNAseq assays is required, particularly because the neurogenic trajectory which is enhanced in the Oct4/Rbpj-KO context is also evident in the control retina. Claims regarding the involvement of transcription factors in adult neurogenesis (such as Rfx4) need to be toned down unless they are backed up with functional data. It is possible that such factors are important, but equally, they may have no role or a redundant role, and without functional tests, it's impossible to say one way or the other.Overall, the authors achieved what they set out to do, and have made new insights into how neurogenesis can be stimulated in MG. Ultimately, a major long-term goal in the field is to replace lost photoreceptors as this is most relevant to many human visual disorders, and while this paper (like all others before it) does not generate rods or cones, it opens new strategies to coax MG to form a related neuronal cell type. Their approach underscores the benefits of using a gold-standard approach for lineage tracing.

We thank the Reviewer for their evaluation. We have made extensive changes to the manuscript to correct errors and modify discussion as recommended. These are detailed below in our point-by-point responses to specific recommendations to the authors.

**Recommendations for the authors:**

**Reviewer #1 (Recommendations for the authors):**
Minor corrections:(1) In Figure 1C top GFAP-mCherry panel, two dim GFP + cells have colocalized with Otx2, is it caused by optic imaging thickness or some muller glia cells having the Otx2 expression?

This indeed reflects the effects of optic imaging thickness. Colocalization of Sun1-GFP and Otx2 is not observed when Z-stack images are examined in GlastCreER;Sun1-GFP retinas. This can also be appreciated by the fact that, in cases of apparent overlap of nuclear envelope-targeted Sun1 and Otx2, the sizes of the labeled areas differ. In cases of true expression overlap, such as is seen following Oct4 overexpression, the labeled areas are the same size, or very nearly so.

Whether the Glast-CreERT2 x Rosa26-LSL-Sun1-GFP mouse line has cross-labeling with the Otx2+ bipolar cells, the author should image the mCherry ctrl sample with a thin optical imaging layer with a small pinhole for Z-stack to verify the co-labeling the GFP and Otx2 in mCherry ctrl sample.

Please see above. Since we first described this line (de Melo, et al. 2012), we have examined thousands of sections of GlastCreER;Sun1-GFP retinas, and have yet to see a single GFP-positive neuron. To avoid confusion, however, we have replaced these images with an additional image from a control mCherry-infected GlastCreER;Sun1-GFP retina processed for the same study.

In the middle upper panel, Oct4-mCherry group, the white arrows indicate the GFP colocalized with Otx2 signal, but seems not mCherry positive, by contrast, the neighbor cells have significant mCherry expression but no colocalization with Otx2. The GFAP promoter-Oct4-mCherry may have stopped expression after the Müller Glia cells were converted into Otx2+ bipolar cells, but is there any middle stage in which the Oct4mCherry and Otx2 co-expression? And after Müller glia to Bipolar conversion, why have Glast-CreERT2 driven GFP expressions not suppressed as GFAP promoter driven Oct4-mCherry? Could the author discuss this point?

We observed a significant number of Muller glia-derived cells expressing both Otx2 and weak mCherry signal. GFP expression is driven by the ubiquitous CAG promoter following Cre-dependent excision of a transcriptional stop cassette. We have modified the text to make this point explicit.

(2) In Figure S2b, the mouse is labeled with wild type; I assume it should be the same mouse line as Fig.1. Otherwise, the author should describe the source of the GFP signal.

“Wildtype” in this case refers to GlastCreER;Sun1-GFP controls, which as the Reviewer correctly points out, are not truly wildtype. The genotype of these animals is specified in all figure legends, and is now referred to as “control” rather than “wildtype” in the figures and main text throughout.

In Figure S2k and l, mCherry ctrl panel, the GFP+ cells looked co-labeling with Otx2, so again, is it the thicker optical imaging layer that caused overlapping vertically or the low specific of Müller Glia of the mouse line? Please describe the stars' meaning in Figure S2i,j in the figure legend. There are 2 figures labeled "n" of the quantification data.

This is, again, an example of the thicker optical imaging layer causing apparent overlap. We have previously demonstrated that the Sun1-GFP+ cells do not co-label with Otx2 in GFAP-mCherry AAV-injected control retinas (Le et al., 2022; Fig. 2C). The asterisks (*) indicate mouse-on-mouse vascular staining, which is now clarified in the figure legend. The 2 figures labeled ‘n’ have been relabeled as ‘m’ and ‘n’.

(3) In Figure 2c in the top panel, the Otx2 image was wrong; please replace it with the correct one.

We thank the Reviewer for spotting this error. This is an inadvertent duplication of the single-channel Otx2 staining for mCherry control sample. We have replaced this with the correct image.

(4) In Figure 3a, the Rbpj-cKO mouse line was used, but where was the GFP signal from? Please verify the mouse line you used in your work. The same question is also asked in Figure S3, S4b.

GlastCreER;Rpbj^lox/lox^;Sun1-GFP were used in Figure 3a. As now specified in the Methods and all figure legends, all mice used in this study carry both the GlastCreER and Sun1-GFP transgenes.

(5) In Figure S4c,d, and 5 wks time point, if the authors quantify the GFP+/Sox2- cells changing, it will be more helpful to understand the percentage of the Müller glia cells conversion to Bipolar cells compared to the Figure 2D, and can be as a supplement to the conclusion Müller to Bipolar conversion rather the Müller proliferation.

Sox2-/GFP+ cells are a measure of Müller glia to bipolar cell conversion that complements that of GFP+/Otx2+ cells. This is now clarified in the text. We also include quantification of Sox2-/GFP+ neurons at 5 weeks post-injury in Fig. S5b.

(6) In Figure S1b,c, there is a large portion of cells that are activated Müller glia after NMDA injury. Did the activated Müller glial cells lose their Müller glial identity? Between the loss of Müller glial identity and neuronal reprogramming, are there any markers that can be used to assess whether Müller glial cells are truly transdifferentiating into neurons rather than remaining in a reactive glial state or an intermediate phase?

Wildtype Müller glia progressively revert to resting state, and by 72 hours post-injury have already lost expression of Klf4 and Myc (Hoang, et al. 2020), a point which is now specifically mentioned in the text. In GlastCreER;Sun1-GFP;Nfia/b/x^lox/lox^;Rbpj^lox/lox^ Müller glia, reactive MG appear to largely convert to bipolar and amacrine-like cells, and it remains unclear if they eventually revert to a resting state (Le, et al. 2024).

**Reviewer #2 (Recommendations for the authors):**
This work demonstrates that Oct4 (Pou5f3) can induce neurogenesis in murine Müller glia (MG). Le et al start by showing that murine and zebrafish MG lack expression of Oct4 (Pou5f3) and its target Nanog. To assess the effect of Oct4 they first label adult MG with Sun1-GFP using tamoxifen-treated GlastCreER;Sun1-GFP mice, then later transduce in vivo with AAV vectors expressing mCherry alone or Oct4 + mCherry. Subsequently, they damage the retina with NMDA and assess the effects several weeks later. In Oct4+ cells at 2 weeks there is rare induction of the neural determinant Ascl1, down-regulation of the MG marker Sox2, induction of bipolar markers (Otx2, Scgn,Cabp5) but not amacrine (HuC/D) or rod (Nrl) markers. Combining Oct4 withNotch inhibition (deleting floxed Rbpj) synergistically increases bipolar cell induction, with Otx2 staining rising to >20% of GFP-marked cells, and cells losing MG identify (loss of Sox2/9). EdU labeling was negligible suggesting direct trans-differentiation. Similar synergy was seen upon combining Oct4 expression with Notch1/2 double gene knockout. Attempts to combine Oct4 with Nfia, Nfib, and Nfix loss were unsuccessful as the GFAP promoter driving Oct4 in MG seems to require these three related transcription factors. scRNAseq confirmed the Oct4-overexpression/Rbpj-KO-driven increase in bipolar cells and decrease in MG cells and revealed that these manipulations may enhance bipolar cell genesis by repressing genes that define quiescent MG and enhancing expression of genes that define reactive MG and neurogenic cells. Finally, multiomic snRNA/scATAC-seq data was performed to assess the effect of Oct2 in wt or Rbpj null MG. This approach revealed that, as anticipated, more genes were up and down-regulated in the context of both manipulations vs Oct4 OE alone. Moreover, Oct4 and Rbpj KO reduced chromatin accessibility at target motifs for transcription factors involved in MG identify/quiescence, while MGPCs showed elevated accessibility for neurogenic factors. The combination of Oct4 OE and Rbpj KO induces accessibility at various interesting TF sites that may contribute to the synergistic neurogenesis, including Rfx4, Klf4, Insm1, and others.This is an interesting paper that adds to the growing literature on how neurogenesis can be induced in mammalian MG. The focus on Oct4 is interesting and the synergistic effects are striking and analyzed in some detail with scRNAseq and multiomic snRNA/scATACseq. The latter results provide useful new insight into transcriptional programs that may be critical in driving neurogenesis. Functional insight into these new candidates is not explored in this manuscript, but that's beyond the scope of the current work and forms the basis for new studies. There are some overreaching statements in the Discussion that need to be toned down, but apart from that and a long list of textual errors that need to be fixed, this paper is a valuable contribution to the field.Major commentsThere are numerous textual errors (some, but not all, examples are detailed in minor comments). It was difficult to follow this paper given the unusually high number of textual errors and the abbreviated legends. Greater attention should be paid to harmonizing the text with the figures and ensuring that the legends are correct and complete.

The manuscript has been proofread carefully and errors corrected.

The opening section of the scRNAseq data should outline briefly why sorting for GFP labeled cells purifies a significant fraction of non-MG cell types, despite the earlier claim, (which agrees with other publications), that GLAST-CreER transgene expression is highly specific to MG. Presumably, it mainly/totally reflects the co-purification of cells, cell fragments, and/or cell-free mRNA from other lineages. Is it also possible that a fraction (however small) of these cells reflect low-level spurious/temporary activation of GLAST-CreER expression in non-MG? The "contamination" is present despite the addition of the GFP sequence to the reference genome (as explained in Methods). They mention: "a clear differentiation trajectory connecting Muller glia, neurogenic Muller gliaderived progenitor cells (MGPCs), and differentiating amacrine and bipolar cells (Fig. 3b)". However, the same trajectory is evident in control mCherry samples, so one could argue that this trajectory is active in normal retina at some low rate, but that would/should equate to rare sun-GFP+ non-MG in controls. Are there any such cells, even extremely rarely, or is it truly 0%? At any rate, the authors need to raise these concerns and offer some explanation(s) at the start of their scRNAseq Results section. If there are really no such sun-GFP+ cells, the authors should comment on the presence of the apparent inactive trajectory in the Discussion.

Since we first described this line (de Melo, et al. 2012), we have examined thousands of sections of GlastCreER;Sun1-GFP retinas, and have yet to see a single GFP-positive neuron. We have also previously shown (Hoang, et al. 2020) that FACSbased isolation of GFP-positive cells from GlastCreER;Sun1-GFP yields a roughly thirty-fold enrichment of Muller glia, implying the presence of small numbers of contaminating neurons. We thereby conclude that the presence of small numbers of neurons (rods, cones, bipolar, and amacrine cells) in the control GlastCreER;Sun1-GFP represents contamination rather than low levels of glia-to-neuron conversion, particularly since we are unable to detect the expression of genes such as neurogenic bHLH factors or immature photoreceptor precursor-specific factors such as Prdm1 that indicate the presence of intermediate cell states. This is now addressed in the Results section related to both Figures 3 and 4.

Discussion:In reference to other strategies to induce neurogenesis the authors make the claim that Oct4 is fundamentally different: "In these cases, Müller glia broadly upregulate proneural genes and/or downregulate Notch signaling. Oct4 instead induces expression of the neurogenic transcription factor Rfx4, which is not expressed in developing retina. It is likely that activation of this parallel pathway to neurogenic competence in part accounts for synergistic induction of neurogenesis seen in Rbpj-deficient Müller glia". First, all these strategies, including Oct4, seem to activate bHLH factors, so they have that in common and the authors should note that overlap. More seriously, without functional tests (e.g. KO Rfx4) the authors need to dial back the over-reaching statement that Rfx4 is the fundamental mechanism driving the Oct4 effect. They can certainly suggest that this is one possibility, but equally, Rfx4 may have very little or no effect on neurogenesis, or it could act redundantly with some of the other factors the authors uncovered. It's impossible to know without functional data, so they either need to add the functional data, or hold back on the strong one-sided and overreaching claim.

Since both Rfx4 expression and motif accessibility are selectively observed following Oct4 overexpression, and Rfx4 also has known neurogenic activity, we stand by our conclusion that it is a particularly strong candidate for mediating the neurogenic effects of Oct4 overexpression. However, the Reviewer is correct that in the absence of functional data, speculation about its function should be qualified. We have done this in the revised manuscript.

Minor commentsThis sentence in the Results is confusing: "While expression of neurogenic bHLH factors driven by the Gfap promoter was rapidly silenced in Muller glia and activated in amacrine and retinal ganglion cells, Gfap-Oct4-mCherry remained selectively expressed in Muller glia but did not induce detectable levels of Muller glia-derived neurogenesis in the uninjured retina (Le et al., 2022)". The cited reference is at the end so it sounds like the Oct4 assay was performed in Le et al 2022, and there is no reference to a Figure for the Oct4 data in the current paper.

As stated here, in Le, et al. 2022, we did not observe any conversion of Sun1-GFP-positive Muller glia to neurons in the absence of injury. In the current study, we instead test whether NMDA-induced excitotoxicity induced glia to neuron conversion in Muller glia overexpressing Oct4. This is now made clear in the revised text.

There are many errors and omissions regarding Figure S2:Figure S2a, b legend, and panels do not match. 2a should be a schematic of the strategy to label MG with Sun1-GFP using GLAST-Cre and a floxed Sun1-GFP allele, but that's missing and instead, the current 2a is a schematic of AAV vectors. It seems that the current 2b legend may describe the combination of the current 2a and 2b panels.

This has been corrected.

Figure S2: Asterisks label certain stained elements in the Oct4 labeled panels, but there is no explanation in the legend. Are these meant to indicate non-specific staining? If so, what is the evidence that the signal is non-specific?

These asterisks represent non-specific mouse-on-mouse vascular staining observed with the mouse monoclonal anti-Oct4 used in this study. This is now indicated in the figure legend.

The text refers to Ascl1 staining in Figure S2e,f, but it's S2g,h.

This has been corrected.

Re this: "While Sun1-GFP-positive cells infected with Oct4-mCherry mostly express the Muller glial marker Sox2 (Fig. S2a,b), from 2 weeks post-injury onwards a subset of GFP positive cells did not show detectable Sox2 expression (Fig. S2b, yellow arrows)". Figure S2a, b are schematic diagrams, not immunofluorescence. They probably mean Figure S2c, d.

This has been corrected.

Fig S2m is mislabeled "n".

This has been corrected.

There are probably other errors with this figure, but I mostly gave up at this point. The authors should go through the paper to find and correct any additional mistakes/omissions in the text and legends.

The manuscript has been carefully proofread and errors corrected.

The figure panels are not always mentioned in the order that they appear. There are many examples.

Figure panels are now mentioned in the order that they appear.

Several schematics use "d-18-14" to indicate "day -18 to -14". The former is at first uninterpretable or at best unclear (could mean day -18 to day 14), perhaps d -18 to -14, or d -18:-14 would be clearer.

This has been corrected.

Re: "AAV-infected wildtype Muller glia could be readily identified by selective expression of Oct4 (Fig. 4e). Wildtype Oct4-expressing Muller glia give rise to both small numbers of neurogenic MGPCs (Fig. 4b),". Figure 4E is labeled Pou5f1, but it would be helpful to avoid confusion by also indicating on the figure that Pou5f1 = Oct4; and Fig 4b does not indicate neurogenic MGPCs (perhaps they mean 4c).

This has been corrected.

Some parts of the Results are written in the present tense and should be in the past tense (for guidance: https://www.nature.com/scitable/topicpage/effective-writing-13815989/).

Past tense is now used throughout.

Pit1 (Pou1f1) is referred to as a "close variant" of Oct4/Pou4f5, but this is unclear (e.g. variant could mean a splice variant from the same locus) and the term "paralogue" should be used.

“Paralogue” is now used in this context.

Re: "Infection with Oct4-mCherry vector induced both Oct4 (Fig. S5e) and Ascl1 (Fig. S5d) expression in Notch1/2-deficient Müller glia." Supplementary image 5d is the one depicting Oct4 and 5e is the one showing Ascl1. However, the reference is reversed.

This has been corrected.